# Variation and Driving Factors of Water Discharge and Sediment Load in Different Regions of the Jinsha River Basin in China in the Past 50 Years

**Shang-Wu Liu [1], Xiao-Feng Zhang [1,*], Quan-Xi Xu [2], De-Chun Liu [3], Jing Yuan [2] and Miao-Lin Wang [3]**

1    State Key Laboratory of Water Resources and Hydropower Engineering Science of Wuhan University, Wuhan 430072, China; liusw2017@whu.edu.cn
2    Bureau of Hydrology, Changjiang Water Resource Commission, Wuhan 430010, China; xuqx@cjh.com.cn (Q.-X.X.); yuanj@cjh.com.cn (J.Y.)
3    Upper Changjiang River Bureau of Hydrological and Water Resources Survey, Changjiang Water Resources Commission, Chongqing 400014, China; syliudc@cjh.com.cn (D.-C.L.); sywangml@cjh.com.cn (M.-L.W.)
*    Correspondence: zhangxfwhuee@263.net; Tel.: +86-027-68772215

**Abstract:** The Jinsha River is the main source of sediment in the Yangtze River Basin. The variation of water discharge and sediment load not only affects the operation of the cascade reservoirs in the basin but also change the water and sediment conditions into the Three Gorges Reservoir. The Jinsha River Basin is divided into six regions based on the measured data of hydrological stations. Herein, the variation regularity and driving factors of water discharge and sediment load in the Jinsha River Basin are analyzed in the past 50 years using the Mann–Kendall and Rank Sum Test. Results show that the source of water and sediment in the Jinsha River Basin is different, and the abrupt and trend changes of water discharge and sediment load in different regions are evident different. Changes in precipitation, water and soil conservation projects, and the construction of reservoirs are the main driving factors of sediment load variation. The average annual sediment reduction load in the Jinsha River from 1998 to 2015 is approximately $99.57 \times 10^6$ t/y, of which the contributions of water discharge change and human activities to sediment load are 18.9% and 81.1%, respectively. The reduction of sediment load in the Jinsha River Basin can result in evident decrease in the sedimentation of cascade reservoirs, erosion of the downstream channel of the river, and considerable reduction of sediment load into the Three Gorges Reservoir.

**Keywords:** sediment load; water discharge; water–sediment relationship; driving factor; reservoir construction; effect of sediment reduction

## 1. Introduction

Rivers are major pathways that link continents and oceans and deliver large quantities of land-derived materials (including fresh water, sediment, elements, and nutrients) to the oceans [1,2]. These pathways are highly important to maintain the fluvial geomorphology of rivers, deltas and stabilize the "land-to-sea" ecosystem. Approximately $20 \times 10^9$ t/y of terrigenous materials enter the ocean every year [3]. Sediment and water are the most important substances in rivers, and their transportation is vital to maintain the stability of the dynamic geomorphologic processes in rivers, offshore, and deltas. However, their transport are gradually deteriorating due to global climate change and increased human activities. On this basis, change of the water discharge and sediment load of major rivers have also become a topic of interest worldwide. In the past 60 years, under the background of climate change and the increasingly extensive and intensive human activities, sediment load in many

rivers worldwide has changed remarkably. Global warming accelerates the hydrological cycle, which changes the temporal and spatial characteristics of precipitation, thereby leading to frequent extreme flood events and affecting the sediment transport capacity in rivers. Large-scale human activities, such as soil and water conservation projects, construction of large reservoirs, and the changes of land use types, also affect sediment transport in rivers [4–6]. Milliman [7] showed that the sediment load in many European rivers has decreased considerably in the past 50 years. Studying the long-term data of 145 rivers in Asia, Europe, and North America, Walling et al. [8] found that approximately 50% of the sediment load in rivers is changing, of which the majority is going downward. Based on the sediment data of 133 hydrological stations in the Tropical River Basin in India from 1986 to 2005, Panda et al. [9] found that the sediment load of approximately 88% of the hydrological stations exhibit a considerable decreasing trend due to global climate change and human activities. Many other rivers in the world, such as the Red River [10], Mississippi River [11], Hanjiang River [12], and Gongshui [13] rivers, also experience decreasing sediment load. However, the variation of water discharge is unobvious in most rivers worldwide [3,13].

The Yangtze River is the third longest river in the world and the longest river in China. The Jinsha River, which flows through Qinghai, Tibet, Sichuan, and Yunnan Province in China, is located in the upper reaches of the Yangtze River [3]. It is also the main source of sediment of the Yangtze River Basin and the Three Gorges Reservoir and the largest hydropower-producing region in China [14,15]. The variations of water discharge and sediment load in the Jinsha River not only greatly influence the operation of the cascade reservoirs in this region but also change the condition of sediment into the Three Gorges Reservoir. Many scholars have conducted studies to explore the change regulation of sediment load in the Jinsha River Basin and obtained numerous important achievements. Zhao et al. [16] concluded that sediment transport in the Jinsha River Basin showed a minimal, notable downward trend based on the sediment data from Pingshan Station from 1956 to 2009. Xiong et al. [17] observed no evident change of trend in sediment load in the Jinsha River basin compared with the sediment data of Pingshan Station from 1991 to 2005 and from 1954 to 1990. Zhang et al. [18] found that the sediment load in the Jinsha River Basin has decreased considerably since the 21st century and that in Pingshan station declined by 34% from 2001 to 2005 compared with that from 1956 to 1970. These researchers used different methods based on different timescale data, and differences in understanding might exist. In recent years, the intensity of human activities in the Jinsha River Basin has increased greatly due to the construction of many cascade reservoirs. Consequently, the sediment load of some hydrological stations in the Jinsha River has decreased dramatically. For example, the sediment load in Panzhihua Station from 2011 to 2015 is 81.5% lower than that from 1966 to 2010, and the sediment load in Pingshan Station from 2013 to 2015 is 99.3% lower than that from 1966 to 1992. However, the variation of water discharge load is unobvious in the Jinsha River [16–18]. Few studies on the changes of water discharge and sediment load transport characteristics under such intense human activities are currently available. In the past, changes of water discharge and sediment load in the Pingshan Station were generally used to reflect the modification of those in the entire Jinsha River Basin, thereby resulting in insufficient comprehensive understanding of the changes of the interior basin. Thus, the current changes of water discharge and sediment load in the Jinsha River Basin and the reason for and extent of the effect of human activities need be recognized and studied.

Therefore, this study is aimed at the analysis of the variation of water discharge and sediment load in the Jinsha River Basin in the past 50 years. We collected the data of water discharge and sediment load and precipitation in the Jinsha River basin, divided the whole basin into six parts by the distribution of hydrological station, and used the method of Mann–Kendall and Rank Sum Test to analyze the change of trend and abrupt point of water discharge and sediment load. The objective of this work were to (1) detect the change characteristic of water discharge and sediment load in different parts of the Jinsha River Basin; (2) analyze the influences of precipitation, water and soil conservation projects, and construction of large reservoirs on sediment load; (3) estimate the contributions of the variation of water discharge and human activities to the variation of sediment load; (4) discuss the

influences of the erosion of the downstream channel and the sediment load into the Three Gorges Reservoir under the background of sediment load reduction in detail. This study could provide some suggestion for the management of cascade reservoirs in the Jinsha River and the Three Gorges Reservoir in addition to facilitating the comprehensive understanding of readers.

## 2. Study Area and Methods

### 2.1. Study Area

The Jinsha River in Qinghai Province originates from the northern foot of the Gladan Winter Snow Mountain in the Tanggula Mountains, which is widely known as the "Third Pole" and "Water Tower of Asia". The area of the Jinsha River Basin (Figure 1) is approximately $500 \times 10^3$ km$^2$, and the total altitude difference reaches 5142 m, accounting for 27.8% and 95% of those of the Yangtze River Basin, respectively. The Jinsha River Basin is divided into three parts by the Shigu and Panzhihua Station, namely the upper, middle, and lower reaches. The lower reaches of the Jinsha River Basin is typical dry–hot valley zone with serious erosion and the main region for water and soil conservation project implementation [19]. The Jinsha River Basin has many tributaries, and the largest one is the Yalong River. The Jinsha River Basin is the largest hydropower-producing region in China. The Ertan Reservoir, which has operated since 1998, is located in the lower reaches of the Yalong River, 33 km away from the junction of the Yalong and Jinsha River. Then, large reservoirs, such as Jinping's first and second Reservoir and the Guandi Reservoir, have been operated successively in the Yalong River since 2005. The middle reaches of the Jinsha River Basin have been developed according to the "one reservoir and eight levels" scheme, which refers to eight reservoirs that play a role as one reservoir. The eight reservoirs are as follows: the Shanghutiaoxia, the Liangjiaren, the Liyuan, the Ahai, the Jinanqiao, the Longkaikou, the Ludila, and the Guanyinyan Reservoir. All the reservoirs have already been operated, except for the Shanghutiaoxia and Liangjiaren Reservoir. The Wudongde and Baihetan Reservoir are under construction, and the Xiangjiaba and Xiluodu Reservoir, which have been operated since 2012 and 2013, respectively, are found in the lower reaches of the Jinsha River. The main large reservoirs are illustrated in Figure 1.

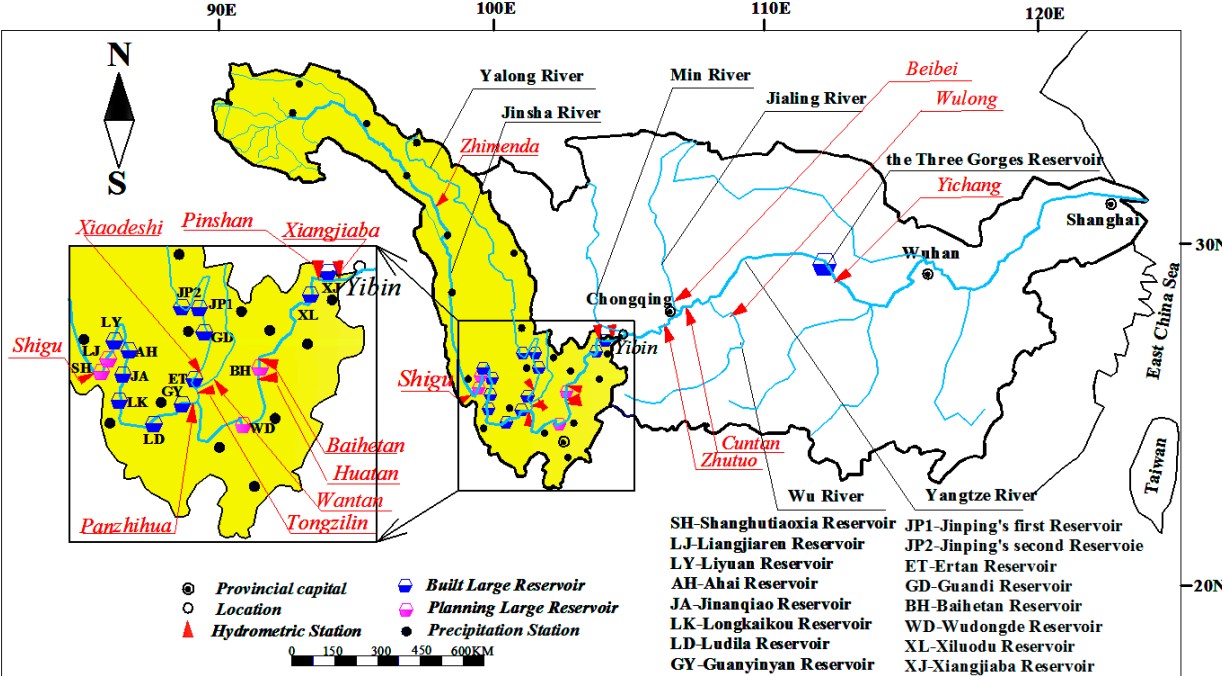

**Figure 1.** Simplified map of the Jinsha River Basin showing the main stream, tributaries, large reservoirs, and hydrological and precipitation stations. The area within the square is enlarged in the inset.

*2.2. Data Series*

In this study, the measured data of water discharge and sediment load of the Zhimenda, Shigu, Huatan, Baihetan, Pingshan, and Xiangjiaba Station of the main Jinsha River, the Tongzilin, Xiaodeshi, and Wantan Station of the Yalong River, and the precipitation stations of the entire basin are collected. The location of the hydrological and precipitation stations is shown in Figure 1. Precipitation data are provided by the National Meteorological Information Center, China Meteorological Administration, and the sediment load and water discharge data are provided by the Hydrological Bureau of Qinghai, Sichuan Province and the Changjiang Water Resource Commission (Table 1).

**Table 1.** Basic information of relevant hydrological stations in the study area.

| Station | River | Catchment Area (km$^2$) | Time Series | Source of Data |
|---|---|---|---|---|
| Zhimenda | Jinsha River | 133,704 | 1960–2015 | Hydrological Bureau of Qinghai province |
| Shigu | Jinsha River | 214,184 | 1952–2015 | Hydrological Bureau of Changjiang Water resource commission |
| Panzhihua | Jinsha River | 259177 | 1966–2015 | Hydrological Bureau of Changjiang Water resource commission |
| Tongzilin | Yalong River | 128,363 | 1999–2015 | Hydrological Bureau of Sichuan province |
| Xiaodeshi | Yalong River | 116,490 | 1963–1998 | Hydrological Bureau of Sichuan province |
| Wantan | Yalong River | 11,100 | 1957–2005 | Hydrological Bureau of Sichuan province |
| Huatan | Jinsha River | 425,949 | 1958–2014 | Hydrological Bureau of Changjiang Water resource commission |
| Baihetan | Jinsha River | 430,308 | 2015 | Hydrological Bureau of Changjiang Water resource commission |
| Pingshan | Jinsha River | 458,592 | 1956–2011 | Hydrological Bureau of Changjiang Water resource commission |
| Xiangjiaba | Jinsha River | 458,800 | 2012–2015 | Hydrological Bureau of Changjiang Water resource commission |
| Zhutuo | Yangtze River | 694,725 | 1954–2015 | Hydrological Bureau of Changjiang Water resource commission |
| Beibei | Jialing River | 156,736 | 1939–2015 | Hydrological Bureau of Changjiang Water resource commission |
| Wulong | Wu River | 83,035 | 1951–2015 | Hydrological Bureau of Changjiang Water resource commission |

In order to discuss the influence of sediment load reduction on the downstream of Xiajiaba Reservoir, the annual average sediment load of Zhutuo Station, Beibei Station and Wulong Station, and the section topography data of reach between Xiangjiaba Station and Yibin City are also provided by Hydrological Bureau of Changjiang Water Resource Commission. The data of sediment particle size distribution are provided by Upper Changjiang River Bureau of Hydrological and Water Resources Survey.

The Tongzilin Station is the export control station of the Yalong River Basin, which has data from 1999 to 2015. Data from the Tongzilin station from 1963 to 1998 are replaced by the sum data of the Wantan and Xiaodeshi Station before the operation of Ertan Reservoir, because the total catchment area of the Xiaodeshi and Wantan Station is only 0.6% different than that of the Tongzilin Station. Xiangjiaba Reservoir was built in 2012, and the Pingshan Station was simultaneously moved to the downstream of the Xiangjiaba Reservoir, namely the Xiangjiaba Station. Data from the Pingshan Station are replaced by that of the Xiangjiaba Station after 2012, because the catchment of the latter is only 0.05% larger than the former. Sediment monitoring in the Huatan Station ceased in 2014. Data from the Huatan Station are replaced by those of the Baihetan Station, because the difference in the catchment area of the two stations is less than 1.0%.

To study the variation characteristics of sediment load in different regions further, the Jinsha River Basin is divided into six parts based on the station distribution (Figure 1). The six parts are as follows: the upstream region of Zhimenda Station, the region between the Zhimenda and Shigu Station, the region between the Shigu and Panzhihua Station, the Yalong River Basin, the region between the Panzhihua and Huatan Station (excluding the Yalong River Basin), and the region between the Huatan and Pingshan Station. Except the upstream region of Zhimenda Station and Yalong River, the water discharge and sediment load in each region are the differences of the measured values of each station

between the upstream and downstream, and that in the upstream region of the Zhimenda Station and Yalong River Basin is directly used by the Zhimenda and Tongzilin Station, respectively.

The study period from 1966 to 2015 is unified to ensure the consistency of data and results, because data of the stations have different time scales.

*2.3. Method*

2.3.1. Mann–Kendall Method

The MK method [20,21] effectively extracts the trend and abrupt change points of data; it is also a nonparametric test without certain distribution. This method is used extensively to analyze the time series of hydrology, temperature, and water quality [12,22,23]. In this study, the Mann–Kendall method was used to detect the trend and abrupt change points of water discharge and sediment load under a significant level condition, $\alpha = 0.05$.

(1) Trend Analysis

The null hypothesis $(H_0)$ states that no significant trend occurs for an independent and identically distributed time series $(x = x_1, x_2, \cdots, x_n)$. The alternative hypothesis $(H_1)$ states that a monotonic trend occurs in $X$. Statistics $S$ is defined as:

$$S = \sum_{i=1}^{k} \sum_{k=i+1}^{n} sgn(x_k - x_i),$$ (1)

where $n$ is the number of data points, and $x_k$ and $x_i$ are the data values in time series and $k$ and $i$ $(k > i)$, respectively. In Equation (1), $sgn(x_k - x_i)$ is the sign function, which is expressed as follows:

$$sgn(x_k - x_i) = \begin{cases} 1, & x_k > x_i \\ 0, & x_k = x_i, \\ -1, & x_k < x_i \end{cases}$$ (2)

$$Var(S) = \frac{n(n-1)(2n+5) - \sum_{p=1}^{q} t_p(p-1)(2p+5)}{18},$$ (3)

where $n$ is the number of data points, $q$ is the group number of samples, and $t_p$ is the number of data of each group. The standardized test statistics $(Z_c)$ is computed as follows:

$$Z_c = \begin{cases} \frac{S-1}{\sqrt{Var(s)}}, & S > 0 \\ 0, & S = 0, \\ \frac{S+1}{\sqrt{Var(s)}}, & S < 0 \end{cases}$$ (4)

$$\beta = Median\left(\frac{x_i - x_j}{i - j}\right), j < i,$$ (5)

where $\beta$ represents the slope; a positive slope represents an increasing trend, and a negative slope represents a decreasing trend. The null hypothesis of no trend $(H_0)$ is rejected at the significant level of $\alpha$ if $|Z| < Z_{\alpha/2}$, where $\alpha$ is the significant level of the test, and $Z_{\alpha/2}$ is the critical value of the standard normal distribution with a probability exceeding $\alpha/2$.

(2) Abrupt Change Analysis

For time series $(x_1, x_2, \cdots, x_n)$, a sequence is constructed as follows:

$$S_k = \sum_{i=1}^{k} \sum_{j=1}^{i-1} \alpha_{ij}, \tag{6}$$

$$a_{ij} = \begin{cases} 1, & x_i > x_j \\ 0, & x_i \leq x_j \end{cases} \quad (1 \leq j \leq i) \tag{7}$$

The statistical variables are defined as follows:

$$UF_k = \frac{|S_k - E(S_k)|}{\sqrt{Var(S_k)}}, \tag{8}$$

$$E(S_k) = \frac{k(k+1)}{4}, \tag{9}$$

$$Var(S_k) = \frac{k(k-1)(2k+5)}{72}, \tag{10}$$

where $UF_k$ follows the standard normal distribution and constitutes a forward sequence curve. The same procedure is followed using the retrograde time series to calculate the backward sequence of statistic $UB_k$. Two broken lines can be easily obtained by plotting the series of $UF_k$ and $UB_k$ in one graph. If an intersection point of the sequence lines of $UF_k$ and $UB_k$ exists between the critical line of $\pm U_{\alpha/2}$, the corresponding time point of the intersection is the abrupt change point.

### 2.3.2. Rank Sum Test

In order to ensure the accuracy of result of the abrupt change analysis, the method of Rank Sum Test [24] is used to identify the significance of abrupt change points in this study. Data are ranked from high to low from the beginning. The number corresponding to each datum is the "rank." When the rank is the same as the data number, the average is calculated. The capacities of the data are defined as $n_1$ and $n_2$, and the sum of the ranks of small capacity is the statistics $W$. When $n_1$ and $n_2 > 10$, $W$ approximates a normal distribution.

$$W \sim N\left( \frac{n_1(n_1 + n_2 + 1)}{2}, \frac{n_1 n_2(n_1 + n_2 + 1)}{12} \right) \tag{11}$$

This study selects the $U$ test, and the calculated statistics is as follows:

$$U = \frac{W - \frac{n_1(n_1 + n_2 + 1)}{2}}{\sqrt{\frac{n_1 n_2(n_1 + n_2 + 1)}{12}}} \sim N(0, 1) \tag{12}$$

The distribution functions of the two samples are $F_1(x)$ and $F_2(x)$. The samples before and after abrupt change points come from the same population, that is, $F_1(x) = F_2(x)$. Certain significant level $\alpha$ is given; thus, if $|U| < U_{a/2}$, then the original hypothesis is accepted, and the abrupt change is insignificant; if $|U| > U_{a/2}$, then the original hypothesis is rejected, and the abrupt change is significant.

## 3. Result

### 3.1. Composition of Water and Sediment in the Jinsha River

The Yangtze River has heterogenous characteristics of water and sediment [25]. Table 2 compares the proportion of annual average sediment load and water discharge and the area of each region to

those of the whole Jinsha Basin. Many large reservoirs have been operated since 1998, resulting in even negative sediment load in some regions. Thus, the statistics in Table 2 is unified from 1966 to 1998. The total area of the region between the Panzhihua and Huatan Station and between the Huatan and Pingshan Station only accounts for 15.5% of the total area of the Jinsha River Basin, whereas the sediment load accounts for 63.9%. The modulus of the sediment in the above region, which is the main source of sediment of the Jinsha River Basin, is approximately four times the average value of the entire basin. However, the main water discharge source of the Jinsha River Basin is the Yalong River.

**Table 2.** Composition results of water and sediment area in each section of the Jinsha River (1966–1998).

| Region | Region Area/km$^2$ | Proportion of Region Area/% | Sediment Load | | Water Discharge | | Modulus of Sediment Load/(t/km$^2$) | Modulus of Water Discharge/(10$^3$ m$^3$/km$^2$) |
| --- | --- | --- | --- | --- | --- | --- | --- | --- |
| | | | /10$^6$ t | /% | /10$^9$ m$^3$ | /% | | |
| Upstream of the Zhimenda Station | 133,704 | 29.2 | 7.91 | 3.0 | 12.1 | 8.6 | 59 | 90 |
| Between the Zhimenda and ShiguStation | 80,480 | 17.5 | 14.39 | 5.6 | 28.8 | 20.5 | 179 | 358 |
| Between the Shigu and PanzhihuaStation | 44,993 | 9.8 | 26.05 | 10.1 | 13.6 | 9.7 | 579 | 302 |
| Yalong River Basin | 128,363 | 28.0 | 44.97 | 17.4 | 58.1 | 41.4 | 350 | 453 |
| Between the Panzhihua and Huatan Station (excluding Yalong River) | 38,409 | 8.4 | 94.55 | 36.5 | 10.0 | 7.1 | 2462 | 260 |
| Between the Huatan and PingshanStation | 32,614 | 7.1 | 71.11 | 27.4 | 17.8 | 12.7 | 2180 | 546 |
| The entire Jinsha River Basin | 458,563 | 100 | 258.98 | 100 | 140.4 | 100 | 565 | 306 |

The differences in underlying surface (the condition of geology, vegetation, soil and so on) and the uneven distribution of precipitation are the main reasons for water and sediment heterogeneity. Sediment yield is minimal in the upstream region of the Zhimenda Station with abundant natural vegetation, little precipitation, and few human activities, because the sediment mainly comes from alpine frozen weathering and the collapse and landslides of valley slopes [26]. Dense virgin forests and abundant precipitation exist in the Yalong River Basin; thus, the appearance of soil erosion is insignificant [27]. However, the region between the Panzhihua and Pingshan Station in the lower reaches of the Jinsha River Basin is a typical dry–hot valley zone with heavy precipitation and movement of intense neotectonic development of evident fault, soft and fragmented rock strata, steep terrain, and difficult vegetation growth. This phenomenon results in frequent gravity erosion, such as debris flow [28,29], making this region the main source of sediment of the Jinsha River Basin.

### 3.2. Trend Change of Water Discharge and Sediment Load

The sediment load in the Yangtze River Basin shows a downward trend, whereas water discharge has not undergone a remarkable change in recent years [16]. The variation characteristics of water discharge and sediment load in different regions of the Jinsha River Basin are obviously different. Except for the upstream region of the Zhimenda Station and the region between the Huatan and Pingshan Station, no obvious trend change of water discharge occurs in the Jinsha River (Table 3). The water discharge in the upstream region of the Zhimenda Station shows an increasing trend, which is mainly caused by the melting glaciers caused by the increase in precipitation and warm climate [17,30]. Meanwhile, the water discharge shows a downward trend, which is mainly caused by the decrease in precipitation in the region between the Huatan and Pingshan Station [31].

The annual average sediment load shows a greatly decreasing trend in the regions between the Shigu and Panzhihua Station, between the Panzhihua and Huatan Station, between the Huatan and

Pingshan Station and the Yalong River Basin, and the annual average sediment load reduction is $0.434 \times 10^6$ t/y, $0.679 \times 10^6$ t/y, $1.816 \times 10^6$ t/y, and $0.669 \times 10^6$ t/y, respectively. By contrast, the annual sediment average load shows an increasing trend in the upstream region of the Zhimenda Station and between the Zhimenda and Shigu Station, and the average annual increment is $0.066 \times 10^6$ t/y and $0.246 \times 10^6$ t/y, respectively.

**Table 3.** Spatial and temporal trend change of the water discharge and sediment load in different regions of the Jinsha River Basin using the Mann–Kendall test (1966–2015).

| Regions | Water Discharge | | Sediment Load | | $Z_{\alpha/2}$ |
|---|---|---|---|---|---|
| | $|Z_c|$ | β | $|Z_c|$ | β | |
| Upstream of Zhimenda | 2.112 | 0.084 | 1.087 | 0.066 | |
| Between the Zhimenda and Shigu Station | 0.393 | 0.014 | 2.802 | 0.246 | |
| Between the Shigu and Panzhihua Station | 0.719 | 0.029 | 2.101 | −0.434 | |
| Yalong River Basin | 0.686 | 0.049 | 3.547 | −0.669 | 1.96 |
| Between thePanzhihua and HuatanStation (excluding the Yalong River) | 0.293 | −0.001 | 1.455 | −0.679 | |
| Between Huatan and Pingshan | 2.710 | −0.144 | 4.134 | −1.861 | |

*3.3. Abrupt Change of Discharge and Sediment Load*

When the significant level is α = 0.05, no significant abrupt change in annual average water discharge in each region of the Jinsha River Basin is observed by the method of Mann–Kendall method and the Rank Sum Test. However, several abrupt change points of annual average sediment load occur in all regions of the Jinsha River Basin, except in the upstream region of the Zhimenda Station. Given the influence of nature and human activities on sediment load, the occurrences of abrupt change points are different, but they all occur after 1998. The statistics of abrupt change points and annual average water discharge and sediment load in different periods of each region are shown in Table 4. The variation processes of water discharge and sediment load from 1966 to 2015 are shown in Figure 2.

**Table 4.** Analysis results of water discharge and sediment load in each region at different periods.

| Region | Abrupt Change Point | | Time/year | Sediment Load/$10^6$ t | Water Discharge/$10^9$ m$^3$ |
|---|---|---|---|---|---|
| | Water | Sediment | | | |
| Upstream region of the Zhimenda Station | — | — | 1966–2015 | 8.7 | 13.1 |
| Between the Zhimenda and ShiguStation | — | 1998 | 1966–2015 | 16.52 | 28.7 |
| | | | 1966–1997 | 13.13 | 28.4 |
| | | | 1998–2015 | 22.56 | 29.2 |
| Between the Shigu and PanzhihuaStations | — | 1998, 2011 | 1966–2015 | 21.80 | 14.7 |
| | | | 1966–1997 | 24.85 | 13.6 |
| | | | 1998–2010 | 29.11 | 18.1 |
| | | | 2011–2015 | −16.69 | 11.7 |
| Yalong River Basin | — | 1999 | 1966–2015 | 34.17 | 58.0 |
| | | | 1966–1998 | 44.97 | 57.5 |
| | | | 1999–2015 | 12.56 | 59.0 |
| Between the Panzhihua and HuatanStation (excluding Yalong River) | — | 2003 | 1966–2015 | 84.24 | 10.1 |
| | | | 1966–2002 | 93.95 | 11.0 |
| | | | 2003–2015 | 56.60 | 7.2 |
| Between the Huatan and Pingshan station | — | 2001, 2013 | 1966–2015 | 54.03 | 17.0 |
| | | | 1966–2000 | 72.58 | 18.1 |
| | | | 2001–2012 | 30.83 | 14.5 |
| | | | 2013–2015 | −68.59 | 13.0 |

The region between the Zhimenda and Shigu Station has an upward abrupt change point in 1998, and the annual average sediment load increases by 71.9% after 1998. Preliminary investigation showed that the construction of highways in this region entered the peak period since 1998, and substantial

abandoned sediment entered the river, resulting in a considerable increase in sediment load. Since then, sediment load has gradually declined with the progressive completion of highways. The region between the Shigu and Panzhihua Station exhibits an upward abrupt change point in 1998, and the annual average sediment load increases by 17.1% from 1998 to 2010 compared with that from 1966 to 1997. The driving factors were identical to those of the region between the Zhimenda and Shigu Station at this period. However, the downward abrupt change point in 2011 is mainly caused by the retention effect of reservoirs, such as the Jinanqiao Reservoir. The Yalong River Basin exhibits a downward abrupt change point in 1999. The operation of the Ertan Reservoir was the main reason for such abrupt change occurrence and has decreased the annual average sediment load by up to 70.1% since 1999 [32]. The region between the Panzhihua and Huatan Station (excluding the Yalong River) exhibits a downward abrupt change point in 2003, and the annual average sediment load from 2003 to 2015 is 39.8% less than that before 2003 because of the decreasing precipitation, soil and water conservation projects, and other human activities. According to statistics, the annual average precipitation in this region from 2003 to 2015 was 67 mm less than that from 1966 to 2002. The region between the Huatan and Pingshan Station has two downward abrupt change points in 2001 and 2013, respectively. The reason for the abrupt change in 2001 was the same as that in the region between the Panzhihua and Huatan Station, whereas that for the abrupt change in 2013 was the operation of the Xiluodu and Xiangjiaba Reservoir [33].

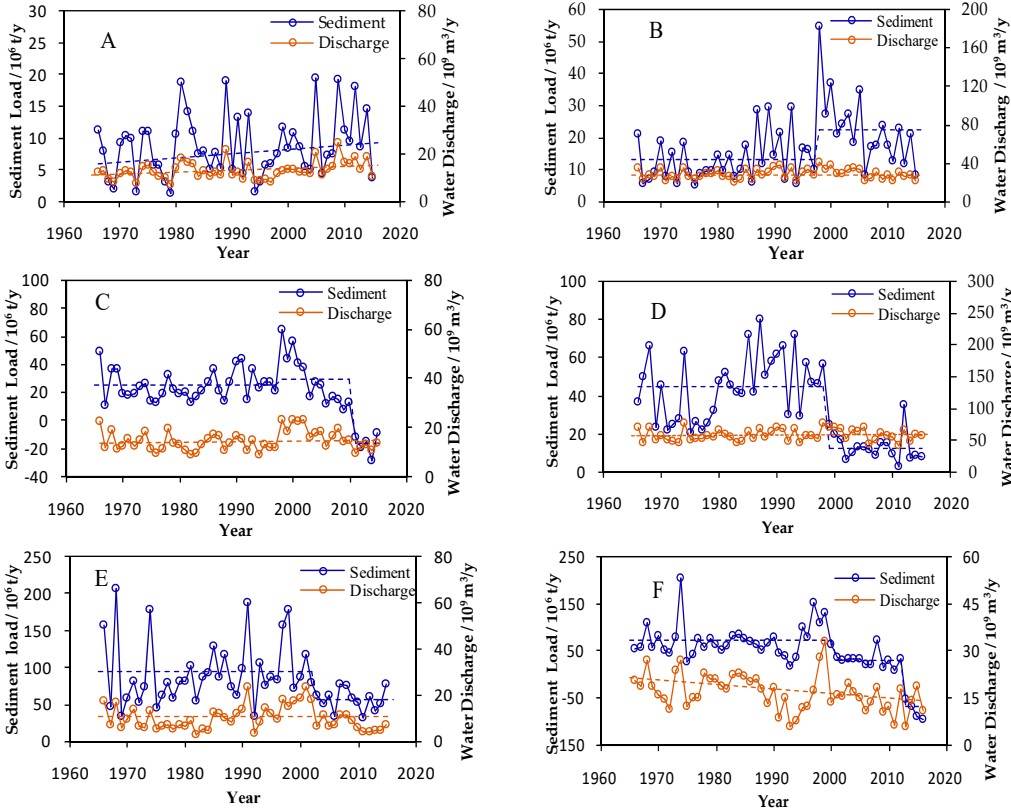

**Figure 2.** Results of variation in water discharge and sediment load in different regions of the Jinsha River Basin from 1966 to 2015 (**A**: Upstream region of the Zhimenda Station; **B**: Region between the Zhimenda and Shigu Station; **C**: Region between the Shigu and Panzhihua Station; **D**: Yalong River Basin; **E**: Region between the Panzhihua and Huatan Station (excluding the Yalong River basin); **F**: Region between the Huatan and Pingshan Station).

### 3.4. Relationship between Water Discharge and Sediment Load

The variations of the relationship between water discharge and sediment load not only reflect the sediment transport capacity in rivers but also the influences of the underlying surface and human

activities on sediment yield in a certain region [34–36]. Such a relationship is often used as a basis for the quantitative estimation of the influences of nature and human activities on the variation of sediment load in rivers [37,38]. This relationship can be generally expressed as a power function as follows:

$$S = aQ^b, \tag{13}$$

where $S$ and $Q$ are the annual average sediment load and water discharge, respectively; and $a$ and $b$ are the coefficients and indices, respectively.

The relationships between water discharge and sediment load (Figure 3) which are divided into several periods by the occurrence time of abrupt change points of annual average sediment load in each region. Either the variation of water discharge or human activities may cause the variation of the relationship. However, the change of sediment load caused by water discharge change is consistent with that of water discharge. So, the variation of water discharge will not cause the systematic deviation. Therefore, the scattered points of the relationship between water discharge and sediment load are mainly caused by human activities. The relationship in the upstream region of the Zhimenda Station is concentrated, and the influence of human activities on sediment load variation is minimal in this region. The relationships are gradually scattered in other regions with the gradual increase in human activities, especially the operation of large reservoirs, such as the Jinanqiao, Ertan, and Xiangjiaba Reservoir in 2011, 1998, and 2012, respectively. Accordingly, the annual average sediment load under the same water discharge evidently decreases compared with the previous period in the region between the Shigu and Panzhihua Station, Yalong River, and between the Huatan and Pingshan Station.

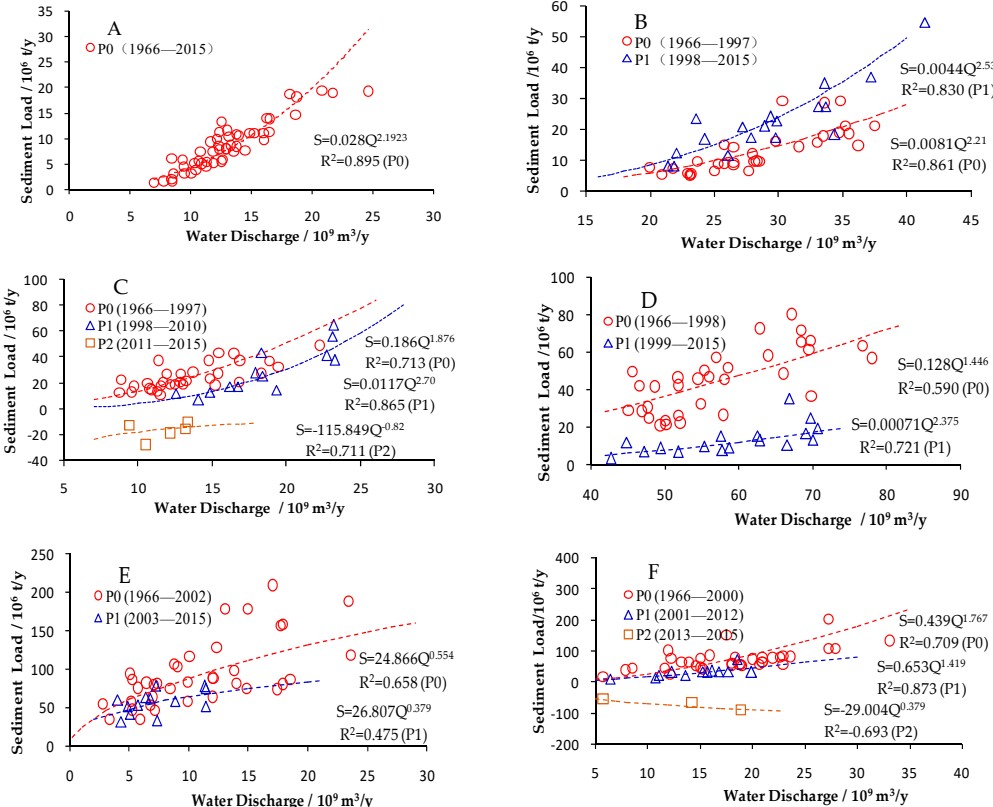

**Figure 3.** Relationship between water discharge and sediment load in each region of the Jinsha River Basin (P0: the basic period; P1: the first measure phase; P2: the second measure phase); **A**: Upstream region of the Zhimenda Station; **B**: Region between the Zhimenda and Shigu Station; **C**: Region between the Shigu and Panzhihua Station; **D**: Yalong River Basin; **E**: Region between the Panzhihua and Huatan Station (excluding the Yalong River basin); **F**: Region between the Huatan and Pingshan Station.

## 4. Analysis and Discussion

### 4.1. Main Driving Factors of Sediment Load

The change of climate, the underlying surface, and human activities are the three main factors that affect the sediment load in a certain basin. The factor of underlying surface includes geological condition, vegetation, soil, and other factors, whereas human activities mainly include deforestation and reclamation, soil and water conservation, and construction of reservoirs. Among these factors, the construction of reservoirs plays a great role in changing the sediment load [39,40]. The underlying surface is relatively fixed within a short time with fewer changes compared with human activities. The main reasons of sediment load variation in the Jinsha River Basin are analyzed as follows:

#### 4.1.1. Precipitation

The change of precipitation is an important factor that affects sediment load variation [41,42]. On the one hand, the processes of precipitation determine the water discharge and affect the variation of sediment load in a certain basin indirectly. On the other hand, the maximum sediment concentration in a river is often synchronized with rainstorm activities and occurrences of gravity erosion, such as landslides and debris flow, which are also related to the intensity and center of a rainstorm [43]. As expected, previous research has shown a positive correlation among precipitation, water discharge, and sediment load [23,44]. Dai et al. [45] concluded that the annual average sediment load in the Pingshan Station from 1993 to 2002 increased by 29% compared with that from 1956 to 1996 due to the increase in precipitation. The region between the Huatan and Pingshan Station is the main sediment source of the Jinsha River Basin and characterized by the serious gravity erosion of landslides and debris. The sediment load from July to September in 2010 in this region increases by $0.015 \times 10^6$ t compared with that in the same period in 2009, with an increase of 118%. Precipitation increases by 118 mm, with an increase of 58.3% in the corresponding time. The annual average precipitation in the upstream region of the Zhimenda Station (Figure 4A) and between the Huatan and Pingshan Station (Figure 4B) present increasing and decreasing trend, respectively, which are both consistent with the variation of the annual average sediment load.

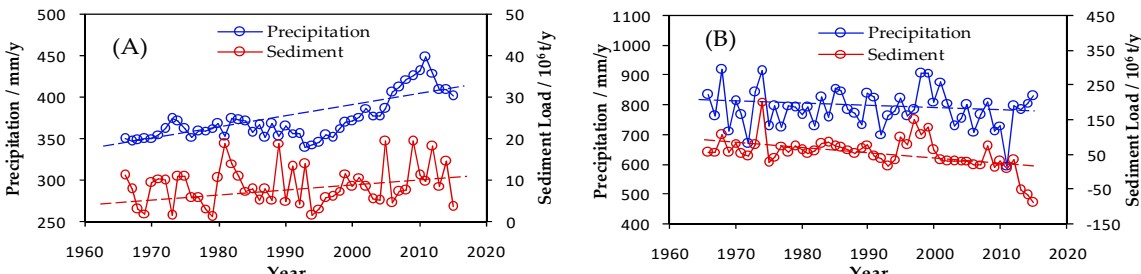

**Figure 4.** Comparison of (**A**) annual average precipitation and (**B**) sediment load from 1966 to 2015.

#### 4.1.2. Human Activities

(1) Water and Soil Conservation

The lower reaches of the Jinsha River Basin experiences the most serious soil erosions in the Yangtze River Basin and is a key region to implement water and soil conservation projects [28]. A total of 400 landslides that are larger than $10^6$ m$^3$ with an estimated total volume of $0.3 \times 10^9$ m$^3$ have occurred on both sides of the lower reaches of the Jinsha River within a length of 15 km. Meanwhile, the quantities of tributary valley debris flow with a drainage area of more than 0.2 km$^2$ are 438, and the quantities of second-level tributary debris flows are 76, and the quantities of slopes of trunk and tributary flows are 37 [46]. The original forest in this region is seriously damaged with poor vegetation. The water and soil conservation project in the Yangtze River began in 1989 and was mainly

implemented in small watersheds. Since the implementation of this project, the area of soil erosion control in the Jinsha River Basin has been $12.3 \times 10^3$ km$^2$ until 2006 [47]. According to the latest investigation of the "Project of Water and Soil Conservation" from 1991 to 2005, the annual average sediment reduction was approximately $(9.6–14.6) \times 10^6$ t/y, with a small benefit of only approximately 4.9% [48]. The effect on sediment reduction is minimal, because the treatment area only accounts for 7.7% of the total area of water and soil loss in this region. Meanwhile, the benefit of the slope control method, which is the major preference for treatment, is ineffective for gravity erosion, such as debris flow and landslide. Notably, the water and soil conservation project need sufficient time to achieve sediment reduction. Therefore, its effect could be insignificant in the early years of implementation, which is one of the reasons why the sediment load in the lower reaches of the Jinsha River Basin did not decrease immediately in 1989.

(2) Construction of Large Reservoirs

Basic Situation of Reservoir Construction in the Jinsha River Basin:

Table 5 shows the basic situation of reservoir construction in the Jinsha River Basin, which is dominated by small or medium reservoirs before 1990 and by large ones after 1991. Statistics show that the capacity of large reservoirs from 1991 to 2005 and from 2006 to 2015 accounts for 93% and 97% of the total reservoir capacity of the same period, respectively. Reservoirs with storage capacities larger than $1.0 \times 10^9$ m$^3$ include the Ertan Reservoir (built in 1998), Jinping's first reservoir (built in 2014), and the cascade reservoirs in the middle and lower reaches of the main stream of the Jinsha River (built since 2010).

**Table 5.** Statistics of reservoirs built in the Jinsha River Basin from 1966 to 2015.

| Period | Large | | Medium | | Small | | Total | |
|---|---|---|---|---|---|---|---|---|
| | Number | Capacity/$10^9$m$^3$ | Number | Capacity/$10^9$m$^3$ | Number | Capacity/$10^9$m$^3$ | Number | Capacity/$10^9$m$^3$ |
| 1966–1990 | 2 | 0.707 | 22 | 0.365 | 1201 | 0.685 | 1225 | 1.757 |
| 1991–2005 | 6 | 7.826 | 21 | 0.425 | 336 | 0.211 | 363 | 8.462 |
| 2006–2015 | 14 | 34.36 | 39 | 1.036 | 166 | 0.172 | 219 | 35.568 |
| 1966–2015 | 22 | 42.893 | 82 | 1.826 | 1703 | 1.068 | 1807 | 45.787 |

Note: Large reservoirs refer to reservoir with storage capacity greater than $0.1 \times 10^9$ m$^3$, and medium reservoirs refer to reservoir with storage capacity between $0.01 \times 10^9$ m$^3$ and $0.1 \times 10^9$ m$^3$, and small reservoirs refer to reservoir with storage capacity less than $0.01 \times 10^9$ m$^3$.

Estimation of Sediment Retention in Major Large Reservoirs:

Large reservoirs are the decisive reason for the reduction in sediment load in most rivers worldwide [39]. Yang et al. [33] concluded that the sediment load in the Jinsha River Basin has substantially decreased since the operation of the Ertan Reservoir in 1998; the sediment load of the Pingshan Station has decreased to zero since the operation of the Xiangjiaba and Xiluodu Reservoir in 2012 and 2013, respectively. Chen et al. [49] estimated the average annual sediment retention to be $69 \times 10^6$ t/y, with large reservoirs accounting for more than 70%, by collecting the data of large, medium, and small reservoirs in the Jinsha River Basin. Previous analysis shows that the abrupt change points of sediment load in the Jinsha River generally exist in large reservoirs when they were operated, indicating that large reservoirs play a decisive role in sediment retention in the Jinsha River Basin. The annual average sediment load retention by reservoirs can be calculated by Equation (14), and the efficiency of sediment retention by reservoirs can be calculated by Equation (15).

$$S_{Retention} = S_{Ungauged\ area} + S_U - S_D \tag{14}$$

$$\eta = \frac{S_{Retention}}{S_{Ungauged\ area} + S_U} \times 100\% \tag{15}$$

In Equations (14) and (15), $S_{Retention}$ is the annual average sediment load retention by reservoirs; $S_U$ is the annual average sediment load of the control station of import reservoirs; $S_D$ is the annual

average sediment load of control station of export reservoirs; $S_{Ungauged\ area}$ is the annual average sediment load from the region between two stations, and it can be calculated by substituting the water discharge after the operation into the relationship between the water and sediment of the previous period before the operation of the reservoirs; $\eta$ is the efficiency of sediment retention by reservoirs.

*Cascade reservoirs in the middle reaches of the Jinsha River*: The cascade reservoirs in the middle reaches of the Jinsha River Basin started operations in 2011, and their import and export control station are the Shigu and Panzhihua Station. The annual average sediment load of the Shigu Station ($S_U$) is $26.24 \times 10^6$ t, and the sediment load of the Panzhihua Station ($S_D$) is $9.55 \times 10^6$ t/y from 2011 to 2015. The annual average water discharge from the region between the two stations is $11.7 \times 10^9$ m$^3$ in corresponding time, and it is substituted into the relationship of $P1$ in Figure 3C. Then, we can estimate $S_{Ungauged\ area}$ is $8.96 \times 10^6$ t/y. Therefore, $S_{Retention}$ and $\eta$ are $25.65 \times 10^6$ t/y and 72.9% from 2011 to 2015, respectively.

*Cascade reservoirs in the Yalong River Basin:* The cascade reservoirs in the Yalong River started operations in 1998. Their export control station is the Tongzilin Station, and $S_D$ is $12.56 \times 10^6$ t from 1999 to 2015. However, no import control station exists for those reservoirs in this region; thus, $S_U$ is taken as zero. The annual average water discharge is $59.0 \times 10^9$ m$^3$/y in the corresponding time, and it is substituted into the relationship of $P0$ in Figure 3D. $S_{Ungauged\ area}$ is $46.56 \times 10^6$ t/y. Therefore, $S_{Retention}$ and $\eta$ are $33.98 \times 10^6$ t/y and 73.0 % from 1999 to 2015, respectively.

*Cascade reservoirs in the lower reaches of the Jinsha River:* The cascade reservoirs in the lower reaches of the Jinsha River started to be operated in 2012. Their import and export control station are the Huatan and Pingshan Station. The annual average sediment load of the Huatan ($S_U$) and Pingshan ($S_D$) Station is $70.20 \times 10^6$ t/y and $1.61 \times 10^6$ t/y from 2013 to 2015, respectively. The annual average water discharge from the region between the two stations is $13.0 \times 10^9$ m$^3$/y in the corresponding time, and it is substituted into the relationship of $P1$ in Figure 3F. Then, we can estimate $S_{Ungauged\ area}$ is $24.87 \times 10^6$ t/y. Therefore, $S_{Retention}$ and $\eta$ are $93.46 \times 10^6$ t/y and 98.3% from 2013 to 2015, respectively.

Compared with other reservoirs in Mekong River [50], Wu River [51] and Lancang River [52,53], the efficiency of sediment retention by reservoirs in Jinsha River is considerable and remarkable. Some other reservoirs are being constructing, such as Wudongde Reservoir and Baihetan Reservoir, which are expected to further the sediment load in Jinsha River Basin.

(3) Other Factors

Large-scale traffic facility construction, mine development, water conservancy projects, and house and road construction, especially in the middle and lower reaches of the Jinsha River Basin, have led to a large amount of discarded soil that flows into rivers during rainstorms. The total amount of discard reaches up to $150 \times 10^6$ t and increases the annual average sediment load by $45 \times 10^6$ t/y [51]. However, the pace of construction has slowed down in recent years.

The lower reaches of the Jinsha River was urbanized with a great effect on vegetation cover, which resulted in destructive exploitation before 1989. Commercial harvesting of natural forests was forbidden after 1998. The projects of natural forests conservation and conversion of farmland to forests were implemented; hence, the vegetation has been restored, and soil erosion has been reduced to a certain extent gradually.

*4.2. Contributions of Water Discharge Change and Human Activities to Sediment Load*

Human activities may have either decreasing or increasing effects on sediment load in the basin. The construction of reservoirs and soil and water conservation projects play a major role in the decrease in sediment load. By contrast, road and house construction, deforestation, and cultivation play a major role in the increase in sediment load. Completely separating the effects of various activities in the Jinsha River Basin is difficult, considering that collecting relevant data is difficult at present. Accordingly, human activities in this study are studied as a whole.

First, the effects of water discharge and human activities on sediment load are estimated by dividing the study period into (1) the basic period with weak human interference and (2) the measure period with strong human interference. Previous analysis has shown that no evident trend change in water discharge and sediment load was observed in the Jinsha River before 1998, indicating that water discharge changes and human activities had minimal influence on sediment load. After 1998, abrupt change points in sediment load have been observed in all regions of the Jinsha River Basin except in the upstream region of the Zhimenda Station because of the deepening influences of several human activities, such as construction of reservoirs and soil and water conservation projects. Therefore, this study unifies the estimation period from 1998 to 2015. The period before the abrupt change is called the "the basic period" (P0). The periods after abrupt change is divided into "the first measure phase" (P1) and "the second measure phase" (P2) according to the occurrence time of abrupt change point (Figure 3).

Based on the relationship between water discharge and sediment load building in the basic period, the theoretical annual average sediment load in measure period (MSL) can be estimated. Then, the annual average sediment reduction load caused by the change of water discharge in each region can be estimate by subtracting the annual average sediment load in basic period (BSL) from MSL. Last, the annual average of sediment load variation ($\overline{\Delta S}_{Water}$) caused by the variation of water discharge in each region can be estimated by accumulating the variation of sediment load in different measure periods then dividing the total time. The calculation formula is Equation (16). What deserve our attention is that the variation of sediment load in the upstream region of the Zhimenda Station is caused by the variation of water discharge, so Equation (16) can be still used to estimate the variation of sediment load based on the data of annual average sediment load from 1966 to 1998 and from 1999 to 2015.

$$\overline{\Delta S}_{Water} = \frac{\sum\limits_{i=1}^{n} \left(a_0 Q_i^{b_0} - S_0\right)\Delta t_i}{N}, \tag{16}$$

where $a_0$ and $b_0$ are the coefficient and index of the relationship between the sediment and water of the basic period, respectively; $i$ is the sequence number of the measure period ($i$ = 1 or 2); $n$ is the number of measure period; $Q_i$ is the annual average water discharge at each measure period; $\Delta t_i$ is the time of each measure period; $S_0$ is the annual average sediment load in the basic period; and $N$ is the total time from 1998 to 2015 ($N$ = 18). If $\overline{\Delta S}_{Water} < 0$, then the variation in water discharge results in the decrease in sediment load; otherwise, the variation in water discharge leads to the increase in sediment load.

The annual average sediment reduction load caused by human activities in each region can be estimated by subtracting the variation load caused by the variation of water discharge in that period from the difference between the annual average sediment load in that period and that in the previous period. The calculation formula is as follows:

$$\overline{\Delta S}_{human} = \frac{\sum\limits_{i=1}^{n} \left[S_i - S_{i-1} - \left(a_0 Q_i^{b_0} - S_0\right)\right]\Delta t_i}{N}, \tag{17}$$

where $S_i$ is the annual average sediment load of each measure period. The remaining symbols are the same as those in Equation (16).

Table 6 shows the change of sediment load variation in the Jinsha River Basin due to the variation of water discharge and human activities from 1998 to 2015, in which the decreasing load is more than the increasing load. The annual average sediment reduction load is approximately $99.57 \times 10^6$ t/y for the entire basin. The contributions of the changes of water discharge and human activities to the variation of sediment load are 18.9% and 81.1%, respectively.

**Table 6.** Estimations of the variation of sediment load caused by water discharge and human activities.

| Variation of Sediment Load/($10^6$ t/y) | 1998–2015 | | | 1998–2010 | | | 2011–2015 | | |
|---|---|---|---|---|---|---|---|---|---|
| | $\overline{\Delta S}_{Water}$ | $\overline{\Delta S}_{human}$ | Sum | $\overline{\Delta S}_{Water}$ | $\overline{\Delta S}_{human}$ | Sum | $\overline{\Delta S}_{Water}$ | $\overline{\Delta S}_{human}$ | Sum |
| Upstream region of the Zhimenda Station | +3.68 | 0 | +3.68 | +1.69 | 0 | +1.69 | +8.89 | 0 | +8.89 |
| Between the Zhimenda and Shigu Station | +0.90 | +8.52 | +9.42 | +2.57 | +9.74 | +12.31 | −3.88 | +5.83 | +1.95 |
| Between the Shigu and Panzhihua Station | +11.09 | −20.74 | −9.65 | +17.70 | −13.44 | +4.26 | −6.08 | −39.72 | −45.80 |
| Between the Panzhihua and Huatan Station (including the Yalong River) | −13.80 | −44.82 | −58.62 | −5.03 | −50.15 | −55.18 | −36.75 | −30.96 | −67.71 |
| Between the Huatan and Pingshan Station | −20.68 | −23.72 | −44.40 | −17.75 | −14.36 | −32.11 | −28.29 | −48.06 | −76.35 |
| Sum | −18.81 | −80.76 | −99.57 | −0.82 | −68.21 | −69.03 | −66.11 | −112.91 | −179.02 |
| Contribution/% | 18.9 | 81.1 | 100 | 1.2 | 98.8 | 100 | 36.9 | 63.1 | 100 |

In order to study the variation of sediment load from 1966 to 2015 further, the Jinsha River Basin is divided into two parts by Shigu Station (i.e., upstream region of Shigu Station and downstream region of Shigu Station) by us. Making the summation statistics of $\overline{\Delta S}_{Water}$ and $\overline{\Delta S}_{human}$ in above regions based on the results in Table 6, we find the facts that the influence of human activity is still greater than that of water discharge change, generally, in spite of the variation of sediment load differs in different region in above regions. The increase of water discharge and human activities, such as the building of house, have led the increase of sediment load in the upstream region of Shigu Station. And the increase of annual average sediment load by water discharge change and human activities is $4.58 \times 10^6$ t/y and $8.52 \times 10^6$ t/y, and the contributions are 35.0% and 65.0%. By contract, the trend of sediment variation in the downstream region of the Shigu Station is mainly decreasing, the annual average reduction load of sediment by water discharge change and human activities is $23.39 \times 10^6$ t/y and $89.28 \times 10^6$ t/y, and the contributions are 20.8% and 79.2%. Large reservoirs are mainly located in the downstream of Shigu Station, so one of the reasons for huge sediment reduction load in this region is the operation of reservoirs, successively.

Considering that many large reservoirs have been operated since 2011 in the middle and lower reaches of the Jinsha River basin, the entire study period is divided into two periods (i.e., from 1998 to 2010 and from 2011 to 2015) to analyze the variation of sediment load after 1998, further. Results show that the annual average sediment load reduction by human activities from 1998 to 2010 and from 2011 to 2015 are $68.21 \times 10^6$ t/y and $112.91 \times 10^6$ t/y, respectively. Furthermore, the intensity of sediment load reduction in the latter period is approximately 1.7 times that of the former period. Therefore, the operation of large reservoirs in the middle and lower reaches have an evident effect on sediment load reduction.

*4.3. Macroeffect of Sediment Reduction in Watershed*

The effects of sediment load reduction in the Jinsha River Basin are mainly reflected in the sedimentation of cascade reservoirs, the erosion in the downstream channel, and the considerable decrease in sediment load into the Three Gorges Reservoir.

4.3.1. Effect of Cascade Reservoirs on Sedimentation

The operation of the Xiangjiaba and Xiluodu Reservoir started in 2012 and 2013, respectively. Chengdu Engineering Corporation Limited [54] predicted that the annual average sediment load retention by the Xiluodu Reservoir was approximately $208 \times 10^6$ t/y in the first 20 years of its operation based on water discharge and sediment load from 1964 to 1973. Zhongnan Engineering Corporation Limited [55] predicted that the annual average sediment load retention by the Xiangjiaba Reservoir was approximately $41.5 \times 10^6$ t/y in the first 20 years of its operation, considering the effect of the Xiluodu Reservoir. Thus, the total annual average sediment load retention by the two reservoirs

is approximately $250 \times 10^6$ t/y. The earlier estimation in this study shows that the annual average sediment load retention by the Xiangjiaba and Xiluodu Reservoir would be $93.47 \times 10^6$ t from 2013 to 2015, which would be $156 \times 10^6$ t/y less than the predicted value of the two reservoirs. In the corresponding period, the annual average sediment load of the import control station of Huatan is $70.2 \times 10^6$ t/y, which is $104 \times 10^6$ t/y less than the value during the feasibility study period. Therefore, the deposition rate of the Xiangjiaba and Xiluodu Reservoir would be considerably lower than the predicted value due to the notable decrease in sediment in the reservoirs.

### 4.3.2. Effect on Downstream Channel Erosion

First of all, the actual sediment concentration in the downstream region is far away from the state of theoretical sediment transport capacity after the construction of reservoirs [56,57]. Figure 5a shows the result of the particle size distribution of sediment changes at the Pingshan Station in recent years. As shown in the figure, the sediment at Pingshan Station has already been refined due to the retention effect of the reservoirs in the middle reaches of the Jinsha River and the main tributaries before the construction of the Xiangjiaba and Xiluodu Reservoir. The sediment was further refined after the construction of the Xiangjiaba and Xiluodu Reservoir. Accordingly, the percentage of sediment with a particle size of less than 0.062 mm increased from 77.1% between 1988 and 2012 to 96.7% in 2015. The sediment load at Pingshan Station decreased from $222.61 \times 10^6$ t to $0.6 \times 10^6$ t in the corresponding period. Secondly, the water flow regulation of reservoir will also cause the erosion or sedimentation of river channel [58]. The specific regulation is mainly manifested by the method of decrease of flood peak flow, increase of the duration of middle water flow and increase of water flow in dry season. For example, the average water flow of Pingshan Station is $1690 \times 10^9$ m$^3$/s in dry season from 2013 to 2015, which is 10.9% higher than that before the operation of Xiangjiaba Reservoir. Due to the increase of water flow, the capacity of sediment transport of river is enhanced, which leads to the erosion of river channel in dry season. Under the combined effect of sediment load reduction, sediment thinning and adjustment of water flow by reservoirs, the downstream channel of the Jinsha River exhibits a considerable channel erosion phenomenon, which is common in the downstream of the Three Gorges Reservoir [59]. The reach between the Xiangjiaba Reservoir and Yibin City is 29.8 km. Statistics show that the total channel erosion in this reach is $22{,}244 \times 10^3$ m$^3$, and the intensity of erosion is $746 \times 10^3$ m$^3$/km. Most erosion locations are concentrated in the main channel of the river bed. The cross section of the Xiangjiaba is taken as an example (Figure 5b), and the cumulative maximum depth of erosion reaches 7.3 m.

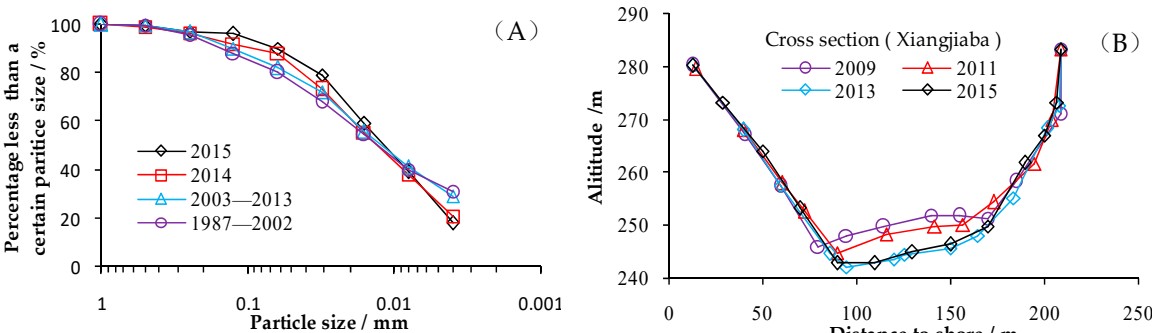

**Figure 5.** Results of downstream channel erosion change in response to sediment reduction (**A**: Changes of sediment particle size in the Pingshan Station; **B**: Changes of topography in the Xiangjiaba cross section).

### 4.3.3. Effect on Sediment into the Three Gorges Reservoir

Presently, the Three Gorges Reservoir is the largest in the world [60]. The sum of sediment load of Zhutuo Station, Wulong Station and Beibei Station in the upper reaches of the Yangtze River can be roughly equal to that of into the Three Gorges Reservoir. According to statistics, the total sediment load

into and out of the Three Gorges Reservoir (Table 7) were approximately $2160 \times 10^6$ t and $520 \times 10^6$ t from 2003 to 2015, respectively. Thus, the total and annual average sediment load retention by the reservoir were $1640 \times 10^6$ t and $126 \times 10^6$ t/y in the corresponding time. However, the actual annual average sediment load retention by the reservoir is only approximately 40% of that in the predicted study [61,62].

**Table 7.** Statistics of sediment load into the Three Gorges Reservoir.

| Period | Annual Average Sediment Load /$10^6$ t/y | | | | |
| --- | --- | --- | --- | --- | --- |
| | Jinsha River (Pingshan Station) | Yangtze River (Zhutuo Station) | Jialing River (Beibei Station) | Wu River (Wulong Station) | Total into the Three Gorges Reservoir (Zhutuo+Beibei+Wulong) |
| Feasibility Study Stage (1961–1970) | 251 | 336 | 179 | 29 | 544 |
| 2003–2012 | 141.63 | 167.86 | 29.15 | 5.7 | 202.71 |
| 2013 | 2.03 | 68.3 | 57.6 | 0.94 | 126.84 |
| 2014 | 2.21 | 34.6 | 14.5 | 6.34 | 55.44 |
| 2015 | 0.6 | 21.2 | 9.54 | 1.28 | 32.02 |

The Jinsha River is the main source of sediment into the Three Gorges Reservoir. During the predicted study stage of the Three Gorges Reservoir, the sediment load of the Jinsha River accounted for 46.1% of the sediment load into the Three Gorges Reservoir. Before the operation of the Xiangjiaba and Xiluodu Reservoir, the annual average sediment into the Three Gorges is approximately $202.71 \times 10^6$ t/y from 2003 to 2012, which is 62.7 % less than that in the predicted study stage. The annual average sediment load from the Pingshan Station is approximately $141.63 \times 10^6$ t/y, which is 43.6% less than that in the predicted study stage from 2003 to 2015. The sediment load from the Jinsha River decreases considerably due to the operation of the Xiangjiaba and Xiluodu Reservoir. The annual average sediment load into the Three Gorges Reservoir decreases by $417.16 \times 10^6$ t/y, $488.56 \times 10^6$ t/y, and $511.98 \times 10^6$ t/y in 2013, 2014, and 2015, respectively, compared with the annual average sediment load of feasibility study stage. The reduction of annual average sediment load from the Jinsha River is $248.97 \times 10^6$ t/y, $248.79 \times 10^6$ t/y, $250.4 \times 10^6$ t/y, and that account for 59.7 %, 50.9 %, 48.9 % of the reduction sediment load into the Three Gorges Reservoir in corresponding time. Thus, the sediment load reduction in the Jinsha River Basin is an important factor for the regulation of sediment load reduction into the Three Gorges Reservoir.

## 5. Conclusions

This study detected the change characteristic of water discharge and sediment load in different parts of the Jinsha River Basin by the method of Mann–Kendall, and analyzed the influences of precipitation, water and soil conservation projects, and construction of large reservoirs on sediment load. Then we estimated the contributions of the variation of water discharge and human activities to the variation of sediment load. Finally, the influences of the erosion of the downstream channel and the sediment load into the Three Gorges Reservoir under the background of sediment load reduction were discussed by us. Some central conclusions can be summarized as follows:

1.　The evident different source of water and sediment in the Jinsha River Basin were caused by the difference in the underlying surface of each region and the uneven distribution of precipitation. The sediment load mainly came from the middle and lower reaches of the Jinsha River, whereas the water discharge mainly came from the Yalong River.

2.　The variation characteristics of the annual average water discharge and sediment load in each region of the Jinsha River Basin were evidently different. The water discharge had an increasing trend in the upstream region of the Zhimanda Station and a decreasing trend in the region between the Huatan and Pingshan Station, but no obvious trend change occurred in the other areas. The sediment load in the upstream region of the Shigu Station showed an increasing trend,

that in the region between the Shigu and Panzhihua Station increased from 1998 to 2010 and then decreased from 2011 to 2015, and in the other regions, it showed a significant decreasing trend.

3. Regarding the change of precipitation, the construction of large reservoirs and soil and water conservation projects, all they were the main driving factors of sediment load reduction in the Jinsha River Basin. The annual average sediment load reduction of the Jinsha River Basin was approximately $99.57 \times 10^6$ t/y from 1998 to 2015. The sediment load reduction values caused by water discharge change and human activities were $18.81 \times 10^6$ t/y and $80.76 \times 10^6$ t/y, which accounted for 18.9 % and 81.1 % of sediment load reduction, respectively.

4. The construction of large reservoirs played a decisive role in sediment load reduction in the Jinsha River Basin. The intensity of sediment reduction load by human activities from 2011 to 2015 was approximately 1.7 times that from 1998 to 2010 mainly due to the operation of large reservoirs in the middle and lower reaches of the Jinsha River since 2011.

5. Under the combined effect of sediment load reduction, sediment thinning and adjustment of water flow by reservoirs, the downstream channel of the Jinsha River exhibited a considerable channel erosion phenomenon. The reduction of sediment load from the Jinsha River Basin, which was the main source of sediment for the Yangtze River, resulted in a considerable reduction of sediment into the Three Gorges Reservoir, thereby affecting the operation of the Three Gorges Reservoir.

**Author Contributions:** S.-W.L., X.-F.Z. contributed to the conception of the study. Q.-X.X., D.C.-L. acquired the data; S.-W.L. contributed significantly to analysis the data and wrote the manuscript. J.Y., M.-L.W. helped revised the manuscript.

**Funding:** This paper was funded by National Key Research and Development Project of China (grant no.2016YFA0600901).

**Acknowledgments:** We are very grateful to anonymous reviewers for their critical reviews of this papers.

**Conflicts of Interest:** The authors declare no conflicts of interest.

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
