# Peer review of "Variation and Driving Factors of Water Discharge and Sediment Load in Different Regions of the Jinsha River Basin in China in the Past 50 Years"

_water, doi:10.3390/w11051109_

Reviewer 1 Report

In my opinion, the research is done well. However, the article is not very interesting. The article is presented not clearly, not interesting. The article should be written more interestingly. This type of research (runoff and sediment load in near to dams or reserviors were described by other Researchers), so the article is not innovative. Of course, probably no one did such analysis in the basin presented by the authors of this work, but the results / dependencies can be transferred / compared with other studies. I do not know where cross-sections are obtained from?

Author Response

     Thanks the suggestion of the review. According to the comments of the review, we improved the paper by the opportunity of this revision. The instruction of significance of this study and modification is as follows:

    The Jinsha River is the main source of sediment of the Yangtze River Basin and the Three Gorges Reservoir and the largest hydropower-producing region in China. This study forced on the change and driving force of water discharge and sediment load of different region in Jinsha River Basin. So, we collected the data of water discharge and sediment load of different stations. Based on the distrubution of hydrological station, the whole basin was divided into six parts. First, we used the method of Mann-Kendall and Sum Rank Test to analyze the trend and abrupt change characteristics of water discharge and sediment load. Then we discussed the influence of precipitation, the project of water and soil conservation and the operation of reservoir on sediment load change. Next, we estimated the contributions of water discharge change and human activities to the variation of sediment load. Last, in order to discuss the influence of sediment reduction on the downstream of the Jinsha River, the annual average sediment load of Zhutuo Station, Beibei Station and Wulong Station, and the section topography data of reach between Xiangjiaba Station and Yibin City were also collected. So, we found the fact that reduction of sediment load in the Jinsha River Basin could result in evident desease in the sedimentation of cascade reservoirs, erosion of the downstream channel of the river, and considerable reduction of sediment into the Three Gorges Reservoir. We want to heip the reader to comprehensive recognize the Jinsha River Basin by our study. Therefore, the study of the sediment reduction load by reservoirs is one but important part in this paper.

    In accordance with the opinions of the review, we make some modification in the corresponding part, as follows:(1) We estimated the efficiency of sediment load reduction by reservoir in Jinsha River Basin in order to compare with other reservoirs in Mekong River, Wu River and Lancang RIver. (2) On the basis of this paper, we discussed the influence discharge adjustment of reservoir on river channel further and compared with other study in the Three Gorges Reservoir.(3) We make some improvment in part of the effect on sediment into the Three Gorges Reservoir. By comparing the sediment load between the Jinsha River Basin and the Three Gorges Reservoir in different period, we found that the sediment load reduction in the Jinsha River is an important for the regulation of sediment load reduction into the Three Gorges Reservoir.

Reviewer 2 Report

Dear authors,

i have read your paper entitled "Variation and driving factors of Runoff and sediment load in different regions of the jinsha river basin in china in the recent 50 years" with high interest. 

It is not common to find such a good amount of data about sediment transport and the possibility to compare them among different years and developing phase of a basin. 

Even if no surprising conclusion is found, luckly, I have appreciated the effort to give a sense and a critical interpretation to such a database.

In my opinion the paper is quite good and should be accepted with some minor revision, mainly on language. English is quite good but some sentences are not clear to me and a bit odd in word order. I suggest some kind of professional proofreading, i will give some suggestion, but I cannot correct the whole article. 

Moreover the use of some words seems inappropriate and not common in hydrology.

For example you refer to discharge rate as runoff. Normally runoff is the fraction of rainfall that does not infiltrate and flows on the slope. So i would prefer Discharge or Flow rate. 

Some key comments by line:

line 39 runoff is not a substance 

line 215 What do you mean by "underlying surface"? it is explained just at the end!

line 218 weathering of alpine frozen? not clear! did you mean weathering of rock due to cold weather?

line 232-234 odd phrase and still don't like runoff load for discharge rate

line 247 why "when alfa=0.05" does results change at different significance levels?

line 304 i like this graphs, well done

line 349: these data are not clear. what is a second level DF?and a slope of a trunk? it is not clear at all

line 417 the lower reaches were urbanized, or economically exploited i guess not developed in economy

line 500 deposition rate soounds better than intensity

line 505 not sure that unsaturated can be used to describe a flow with erosive power.

Conclusions should be improved in the form. 

Point 2: line 555 put some reference years, increased up to and decreased since...

line 557 you cannot start a point say for the changes. Regarding the variation of precipitation... sounds much better

Author Response

    Thanks the suggestion of the review. According to the comments of the reviews, we improved the paper by the opportunity of this revision. The modification instruction of point-by-point is as follows:

line 39  runoff is not a substance

In accodance with the opinion of the review, " Runoff " is replaced by the " Water ".

line 215 What do you mean by " Underlying Surface" ? it is explained just at the end!

" Underlying Surface" refers the surface characteristics of basin, mainly referring to the condition of geology, vegetation and soil. Because of difference in underlying surface, there are evident characteristics of water and sediment heterogeneity in the Jinsha River Basin. Addition explaination have been made in the relenant parts of the paper.

Line 218 weathering of alpine frozen? not clear! did you mean weathering of rock due to the cold weather?

" Weather of alpine frozen" refers to the phenomenon of rock fracturing and collapse in high and middle latituded because of the huge water pressure produced by the process of water freeze in the rock. This part of the broken rocks enter the river and became an important source of sediment.

line 232-234 odd phrase and still don't like runoff load for discharge rate.

In accordance with the opinions of the review, " Runoff load" is replaced by the " Discharge rate".

line 247  why when alfa=0,05 does the results change at different significance levels?

In this paper, the significance level is taken as an uniform value, that is 0.05. In mathematical statistics, " significance level" is a concept of hypothesis test. In the Mann-Kendall analysis, when significance level is 0.05, and the number of samples (water discharge and sediment load in different regions of the Jinsha River Basin) is the same, a unified standard can be used for abrupt and trend change analysis. Because the influence of water discharge and human activites is different in each region, there are different results of abrupt and trend change analysis.

line 304 i like this graphs.

Thanks for your affirmation.

line 349 these data are not clear. What is a second level DF? and a slope of a trunk? it is not clear at all.

" a second level DF" refers to the debris flow large than 10*109m3; " slope debris" is a kind of deris flow. We make some modification in the corresponding part, as follows: the quantities of tributary valley debris flows with a drainage area more than 0.2 km2 is 438, and the quantities of second-level debris flows are 76, and the quantities of slops of trunk and tributary flow are 37.  

line 417 the lower reaches were unbanized, or economically exploited i guess not developed in economy.

In accordance with the opinion of the review, we make some modification in the corresponding part, as follows; The lower reaches of the Jinsha River was unbanized with a great effect on vegetation cover, which resulted in destructive exploitation before 1989.

line 500 deposition rate sounds better than intensity

In accordance with the opinions of the review, " intensity" is replaced by the "rate"

line 505 not sure that unsaturated can be used to describe a flow with erosive power.

" Unsaturated" was used to illustrate that the state of the decrease of sediment concentration under the same  water discharge condition as before the built of reservoirs. We make some modification in the corresponding part, as follows: The actual sediment concentration in the downstream region was far away from the state of theoretical sediment transport capacity. 

Point 2 line 555 put some reference years, increased up to and decrease since

In accordance with the opinion of the review, we put some reference years in the corresponding part: increased from 1998 to 2010 and then decreased from 2011 to 2015.

line 557 you cannot start a point say for the changes. Regarding the variation of precipitation...sounds much better.

In accordance with the opinion of the review, we make some modification in the corresponding part, as follows: Regarding the change of precipitation, the construction of large reservoirs and water and soil conservation project...

In addition, in accordance with the opinion of the review, we have consulted a large number relevant paper, we make some modification in this paper, all " runoff load" are replaced by " water discharge".

Reviewer 3 Report

General observations

The paper reports a study on the runoff and sediment load in the Jinsha River watershed, based on time series of hydrological measures taken at different stations along the basin. Statistical methods were used for the analysis. Different contributions of both climatic and human causes of changes in sediment loads were estimated and discussed. In particular, the effect of reservoirs construction was considered.

The paper appears well organized, despite some sections can be improved. The state-of-the-art and the scope of the paper are clearly presented, as well as the study area, collected data and methods. Results are extensively presented and discussed, and conclusions are consistent with previous sections.

I add below specific observations and suggestions for the different sections of the document.

Overall, I can recommend to accept the paper for publication, provided that the comments and revisions suggested below are considered.

1. Introduction

The introduction is good. The section is concise and synthetically presents the state of the art and an overview of the work.

2. Study area and methods

Please check the label “Pinshan” in the map in Figure 1.

In the Data Series Section, a Table reporting the main information on the measurement stations (e.g. name, parameters, start/end date, etc.) would be appreciated.

Line 167-168: please, check the terms “tired groups” and “tires”. Did you mean “tied”?

Line 177: please, check the summation index j in the equation 6, that seems incomplete.

Line 190: First sentence seems incomplete. Please, check.

3. Results

Table 1: I suggest to specify in the table or in the caption the duration of time series to which the value are referred. I assume that the total amounts of sediment and runoff load in the reference period are reported in the table. Is it correct?

Table 2: As for Table 1, I suggest to indicate the duration of the time interval considered for the analysis.

Table 3 and Figure2: I don’t understand the negative values of sediment load. Are you presenting measured values of sediment loads or variations? Can you please explain?

Line 265: Please , check “1998”, as it seems that “1999” would be correct.

Lines 295-296: The sentence “… is plotted by the abrupt points…” seems not correct or incomplete. Please, check.

Lines 296-298: I would suggest to eliminate the sentence “because of the different ways and degrees of human activities at different periods and regions “ as it seems to anticipate the conclusions.

Lines 298-302: the assumption that points scatter in the graphs in Figure 3 and the presence of human activities are related is not obvious and should be either adequately discussed or presented as an hypothesis.

4. Analysis and discussion

As a general observation, in order to avoid confusion I would suggest to use the measurement unit “t/y” when referred to mean annual sediment load, in this section and everywhere in the document.

Lines 130-131: I would suggest to underline that the positive correlation among precipitation, runoff and sediment load is quite intuitive. So, you could add a sentence like “As expected, previous research…”.

Table 4: How did you classify the reservoirs as “small”, “medium” or “large”? I suggest to add a few lines of explication before the table.

Line 391: If the sentence “Cascade reservoirs in the middle reaches of the Jinsha River:” introduces a sub-section, it should be appropriately formatted (e.g. underlined). The same observation can be applied to lines 398 and 404. Please, check.

Line 423: I would suggest to modify the first sentence as “Human activities may have either…”

Lines 440-449 are difficult to read. Please, consider to reorganize the text. I understand that you basically  assume that the contribution of runoff change can be estimated based on the sediment load estimated with reference to the undisturbed period (P0) formulae, but it could be more clearly be explained.

Lines 440 and 457: I would replace the term “obtained” with “estimated”.

Table 5: I would replace the term “Statistics” with “Estimations” ore something similar.

Lines 467-473 are not clear because you only report in Table 3 the results obtained for the entire period 1998-2015. If you want to keep this supplementary information, I would suggest to add tables with the 1998-2010 and 2011-2015 results.

Lines 474-479: I would suggest to explain more clearly how did you derive the discussed results.

Lines 496-502: It should be emphasized that you are comparing estimations performed at reservoir design stage with your estimations based on sediment and runoff measures. Both are estimated data. You are not presenting measured sedimentation data (obtainable, e.g. by bathymetric measures in the reservoir. Of course you are confident with your results and this is OK, but I would suggest you to use different terms in the discussion. At line 497, for example, I would say “The earlier estimation in this study shows that the annual average sediment load retention by the Xiangjiaba and Xiluodu Reservoir would be…”, and the same at line 501, and so on for the entire section.

Lines 503-521: You present results of sediment grain size analysis and river cross-section measurements that were not described before in the document. A few lines should be added, describing how those data were derived.

Lines  513-521: Bathymetric measures of the channel section indicate an erosion as a consequence of the construction of reservoir. This is certainly correct, but please consider that it is not an obvious result. In fact, despite sediment trapping is a logical consequence of water impoundments and a reduction in sediment availability downstream a dam is to be expected, the actual effects on the channels system depend also on the water flow regulation, that may alternatively induce erosion or sedimentation depending on the velocities regime. I would suggest you to take a look, for example, at the paper: “Brandt, S.A. Classification of geomorphological effects downstream of dams. Catena 2000, 40, 375–401”.

Section 4.3.3: How sediment loads into and out of the Three Gorges Reservoir were measured or estimated? This is a key issue. I would suggest to check and expand the entire section, accurately describe the data sources and discuss more clearly the differences with results derived from your study.

Conclusions

Lines 546-547: The sentence “The obvious characteristics…” is not clear. Please, check.

Line 533: I would suggest to replace the term “obvious” with “evident” or “clear”.

Line 553: Please, check the word Pinshan/Pinshang

Line 557: Please, check “For the changes of precipitation and runoff…” as the sentence is not clear.

Line 567: Please, consider my observations on morphological effects of reservoirs construction.

Kind regards

Author Response

    Thanks the suggestion of the review. According to the comments of the review, we improved the paper by the opportunites of this revision. The instruction of modification of point-by-point is as follows:

The inroduction is good.

Thanks for your affirmation.

please check the label " Pinshan" in the map in Figure 1.

In accordance with the opinion of review, " Pinshan" is replaced by" Pingshan" in Figure 1.

In the Data Series Section, a Table reporting the main information on the measurement station( eg. name, parameters, start/end data,etc.) would be appreciated.

In accordance with the opinion of review, Table 1 is supplemented in order explain the situation of measurement stations clearly.

Line 167-168. Please, check the terms" tired groups" and " tires". Did you mean " tied"?

In this paper, q refers to the group number of samples, and tp refers to the number of data of each group. In accordance with the opinion of the review, we make some modification in the corresponding part, as follows: q is the group number of samples, and tp is the number of data of each group.

line 177. please, check the summation index j in the eqution 6.

In accordance with the opinion of the review, the equation(6) is modified.

line 190. First sentence seems incomplete. Please, check.

In accordance with the opinion of the review, we make some modification in the corresponding part, as follows: In order to ensure the accuracy of result of the abrupt change analysis, the method of Rank Sum Test is used to identify the significance of the abrupt change points in this study.

Table 1: I suggest to specify in the table or in the caption the duration of time series to which the value are referred. I assume that the total amouts of sediment and runoff load in the reference period are reported in the Table, it is correct?

In accordance with the opinion of the review, the duration of time series is supplemented in the Table.

Table 2: As for Table 1, I suggest to indicate the duration of time interval considered for the analysis.

In accordance with the opinion of the review, the duration of time series is supplemented in the Table.

Table 3 and Figure 2: I don' t understand the negative value of sediment load? Are you presenting measured values of sediment load or variation? Can you explain?

For a certain region of the Jinsha River Basin, the annual average sediment load is equal to the difference of sediment load between downstream and upstream control station. When the sediment load is negative, it means that the sediment load of downstream station is less than that of upstream station, and there is the phenomenon of sediment deposition.

Line 265. Please check" 1998", as it seems that "1999" would be correct.

In accordance with the opinion of the review, " 1998" is replaced by "1999".

Line 295-296. The sentence " ...is plotted by the abrupt points..." seems not correct or imcomplete. Please, check.

In accordance with the review, we make some modification in the corresponding part, as follows: The relationship between water discharge and sediment load which is divided into several periods by the occurence time of abrupt change points of annual average sediment load in each region.

line 296-298. I would suggest to eliminate the sentence" because of different ways and degrees of human activities at different period and regions".

In accordance with the opinion of the review, we eliminate this sentence.

Line 298-302. The assumption that points scatter in the graphs in Figure 3 and the presence of human activities are related is not obvious and should be either adequately discussed or presented as an hypothesis.

Either the variation of water discharge or human activities my cause the variation of the relationship between water and sediment. However, the change of sediment load caused by water discharge change is consistent with that of water discharge. That is to say, the variation of water discharge will not cause the systematic deviation and scatter of points. So, points scatter in the graph in Figure 3 and the presence of human activities are related. In accordance with the opinion of the review, we make some modification in the corresponding part.

As a general observation, in order to avoid confusion I would suggest to use the "t/y" when referred to mean average sediment load.

In accordance with the opinion of the review, we use the "t/y" and "m3/y" in order to avoid confusion.

line 130-131. I would to suggest to underline that the positive correlation among precipitation, runoff and sediment in quite intuitive. So,you could add a sentence like " As expected, previous research..."

In accordance with the opinion of the review, we make some modification in the corresponding part, as follows: As expected, previous research has shown a positive correlation among precipitation, water discharge and sediment load.

Table 4. How did you classify the reservoir as " small" "medium" and "large"? I would suggest to add a few lines of explication before the table.

Reservoirs are classified according to their capacity of storage. Large reservoirs refer to reservoir with storage capacity greater than 0.1*109m3, and medium reservoirs refer to reservoir with storage capacity between 0.01*109m3 and 0.1*109m3, and small reservoirs refer to reservoir with storage capacity less than 0.01*109m3. In accordance with the review, the explication is supplemented in Table.

line 391.If the sentence " Cascade reservoirs in the middle reaches of the Jinsha River Basin" introduces a sub-section, it should be qppropriately formatted. The same obsevation can be applied to line 398 and 404.

In accordance with the review, the formatted sentence " Cascade reservoirs in the middle reaches of the Jinsha River"," Cascade reservoirs in the Yalong River" and " Cascade reservoirs in the lower reaches of the Jinsha River " are changed.

line 423.I would suggest to modify the first sentence as " Human activities may have..."

In accordance with the review, we make some modification in the corresponding part, as follows: Human activites may have either decreasing or increasing effects on sediment load.

Line 440-449 are difficult to read. Please, consider to reorganized the text.

In accordance with the review, we make some modification in corresponding part, as follows: Based on the relationship between water discharge and sediment load building in the basic period, the theoretical annual average sediment load in measure period (MSL) can be estimated. Then the annual average sediment reduction caused by the change of water discharge in each region can be estimated by subtracting the annual average sediment load in basic period (BSL) from MSL. Last, the annual average of sediment variation caused by water discharge change in each region can be estimated by accumulating the variation of sediment in different measure periods then dividing the total time.

line 440 and 457, I would replace the term" obtained" with " estimated".

In accordance with the review, the " obtained" is replaced by "estimated".

Table.5 I would replace the term" Statisitcs" with" Estimation".

In accordance with the review, the "Statistics" is replaced by"Estimation".

line 467-473 are not clear because you only report in Table 6 the result obtained for the 1998-2015. If you want to keep this supplementary information, I would suggest to add tables with the 1998-2010 and 2011-2015 results.

The supplementary information of the result of 1998-2010 and 2011-2015 is essential for this paper. This result of the estimation can tell us the fact that the operation of large reservoirs in the middle and lower reaches has an evident effect on sediment load reduction. In accordance with the review, the result sof 1998-2010 and 2011-2015 are supplied in Table.

Line 474-479. I would suggest to explain more clearly how did you derive the discussed results.

In accordance with the review, we make some modification in order to explain more clearly in the corresponding part. The modification can be seen in the text.

line 496-502. I would suggest you to use different terms in the discussion. At line 497, I would say the "...would be ...", and the same as at line 501.

In accordance with the review, we make some modification in the corresponding part.

line 503-521. Your present results of sediment grain size analysis and river cross-section measurements that were not described before in the document. A few line should be add to describe the source of data.

The data of sediment grain size and topography of river cross-section are provided by Upper Changjiang River Bureau of Hydrological and Water Resource Survey. In accordance with opinion, we make some supplement in Section 2.2.

Line 513-521. In fact, despite sediment trapping is a logical consequence of water impoundments and a reduction in sediment availaility downstream a dam is to be expected, the actual effects on the channel system depend also on water flow regulation, that may alterntively induce erosion or sedimentation depending on the velocities regime.

In accordance with the review, we learnt the paper carefully. Based on the view of the paper, we make some modification in the corresponding part. The modification can be seen in the corresponding part of the text.

Section 4.3.3. How sediment load into and out ot the Three Gorges Reservoir were measured or estimated? This a a key issue. I would suggest to check and expand the entire section.

Generally, the sum of sediment load Zhituo Station, Wulong Station and Beibei Station in the upper reaches of the Yangtze River can be roughly equal to that into the Three Gorges Reservoir. We make some supplement of data source in Section 2.2 and more discussion in Section 4.4.3.

Line 546-547. The sentence " The obvious characterisitcs..." in not clear.

In accordance with the review, we make some modification in the corresponding part, as follows: The evident different source of water and sediment in the Jinsha River Basin were caused by the difference in the underlying surface of each region and the uneven distribution of precipitation.

line 553. I would suggest to replace the term " obvious" with" evident".

In accordance with the review, the " obvious" is replace by "evident".

line 553.Check the word Pinshan/Pingshan

In accorance with the review, the "Pinshan" is replaced by" Pingshan".

line 557. Please check the sentence " For the change of precipitation and ..."

In accordance with the review, we make some modification as follows: Regarding the change of precipitation, the construction of large reservoirs and water and soil conservation, all they were the main driving factors of sediment load reduction in the jinsha River.

line 567. Please, consider my observation on morphological effects of reservoirs construction.

In accordance with the review, we make some modification in the corresponding part, as follows: Under the combined effect of sediment load reduction, sediment thinning and adjustment of water flow by reservoir, the downstream channl of Xiangjiaba Reservoir exhibit a considerable channnel erosion phenomenon.

Round  2

Reviewer 1 Report

The paper is corrected. I don't have a new remarks.

This manuscript is a resubmission of an earlier submission. The following is a list of the peer review reports and author responses from that submission.

Round  1

Reviewer 1 Report

The paper is well written and structured, the data analyses are appropriate, the results presented and discussed clearly and the conclusions are based on the processed data. 

However, the paper is rather a technical report than a scientific study, I miss a convincing description of a novel approach applid in the study.

Reviewer 2 Report

I am very sorry but I cannot carry out an extensive and detailed review of the paper as the English language and style must be deeply improved as it is very difficult to understand the paper in its current state. Therefore, I cannot proceed further with my review until an extensive editing of the English language is performed.